# Clinical Outcomes of *Escherichia coli* Acute Bacterial Prostatitis: A Comparative Study of Oral Sequential Therapy with β-Lactam Versus Quinolone Antibiotics

**DOI:** 10.3390/antibiotics14070681

**Published:** 2025-07-05

**Authors:** Laura Gisbert, Beatriz Dietl, Mariona Xercavins, Aina Mateu, María López, Ana Martínez-Urrea, Lucía Boix-Palop, Esther Calbo

**Affiliations:** 1Infectious Diseases Department, Hospital Universitari Mútua Terrassa, 08021 Terrassa, Spain; bdgomezluengo@mutuaterrassa.cat (B.D.); aina.mateusubira@gmail.com (A.M.); anamartinezurrea93@gmail.com (A.M.-U.); lboix@mutuaterrassa.cat (L.B.-P.); ecalbo@mutuaterrassa.es (E.C.); 2Medicine Department, Universitat de Barcelona, 08036 Barcelona, Spain; 3Microbiology Department, CatLab, 08232 Viladecavalls, Spain; 4Infection Control Nursing Team, Hospital Universitari Mútua Terrassa, 08021 Terrassa, Spain; mlopezsanchez@mutuaterrassa.cat; 5Infectious Diseases Department, Universitat Internacional de Catalunya, 08195 Sant Cugat del Vallés, Spain

**Keywords:** acute bacterial prostatitis, ciprofloxacin, cefuroxime

## Abstract

Background/Objectives: Optimal management of acute bacterial prostatitis (ABP) remains uncertain, but the use of antibiotics with good prostatic tissue penetration is critical to prevent recurrence and chronic progression. This study aimed to describe clinical characteristics and outcomes of ABP due to *Escherichia coli* (ABP-*E.coli*), compare effectiveness of sequential high-dose cefuroxime (ABP-CXM) versus ciprofloxacin (ABP-CIP), and identify risk factors for clinical failure. Methods: We conducted a retrospective study including men >18 years diagnosed with ABP-*E. coli* between January 2010 and November 2023 at a 400-bed hospital. Patients received oral cefuroxime (500 mg/8 h) or oral ciprofloxacin (500 mg/12 h). Outcomes over 90 days included clinical cure, recurrence and reinfection. Definitions: Clinical cure—resolution of symptoms without recurrences; recurrence—new ABP episode with the same *E. coli* strain; reinfection—ABP involving different microorganism or *E. coli* strain; clinical failure—lack of cure, recurrence, or reinfection. Results: Among 326 episodes (158 ABP-CXM, 168 ABP-CIP), ABP-CXM patients were younger (median 63.5 vs. 67.5 years, *p* = 0.005) and had fewer comorbidities. Clinical cure was higher in ABP-CIP (96.9% vs. 85.7%, *p* < 0.001). Recurrence occurred only in ABP-CXM (6.96% vs. 0%, *p* < 0.001), while reinfection and mortality were similar. Multivariable analysis showed ciprofloxacin was protective against clinical failure (OR: 0.16, 95% CI: 0.06–0.42, *p* < 0.001), while prior urinary tract infection (UTI) increased failure risk (OR: 2.87, 95% CI: 1.3–6.3). Conclusions: Ciprofloxacin was more effective than cefuroxime in treating ABP-*E. coli.* Patients with recent UTIs may need closer monitoring or alternative therapies.

## 1. Introduction

Acute bacterial prostatitis (ABP) is a common cause of emergency room visits, with an incidence of 2 to 5% in adult males [1], and contributes significantly to antibiotic use [2]. Although rarely life-threatening—except in cases of sepsis [3]—inadequate management increases the risk of chronic prostatitis and recurrent infections [1], negatively impacting quality of life. Recognized as a distinct category of prostatitis by the NIH in 1999 [4], the diagnosis of ABP remains inconsistent, and treatment approaches are heterogenous [5].

Several factors influence the risk of recurrence, abscess formation, or progression to chronic prostatitis [1,6], with adequate antibiotic penetration into prostatic tissue being a critical determinant [7,8]. Fluroquinolones remain the first-line treatment according to current clinical guidelines [9,10] and expert reviews [4,11,12]; however, their use is increasingly limited by rising resistance among *Enterobacterales*.

According to ECDC surveillance reports, fluoroquinolone resistance in invasive *E. coli* isolates in Spain remained consistently high between 2010 and 2023, ranging from 23% to 32%. Although cefuroxime is not specifically reported, resistance to third-generation cephalosporins in *E. coli*—often used as a proxy—ranged from 12.1% to 16.8% during the same period [13]. In clinical practice, ABP often allows early transition to oral therapy, with beta-lactam agents such as cefuroxime frequently prescribed at discharge due to the high prevalence of fluoroquinolone resistance.

We conducted a retrospective study to evaluate whether cefuroxime—the most frequently prescribed oral empirical antibiotic for ABP in our setting—is comparable in effectiveness to guidelines-recommended treatments. The study focused on ABP caused by *E. coli*, the most common urinary pathogen.

Our objective was to compare the clinical characteristics, diagnosis, and outcomes of ABP-*E. coli*, treated with high-dose cefuroxime (ABP-CXM) versus ciprofloxacin (ABP-CIP) in men over 18 years of age, and to identify prognostic factors associated with clinical failure.

## 2. Results

We included 326 episodes of *E. coli* ABP: 158 were treated with cefuroxime and 168 with ciprofloxacin. The eligible individuals’ identification process is shown in Figure 1.

Patients in the ABP-CXM group were younger (63.5 (4–73) vs. 67.5 (58–76); *p* = 0.005) and had fewer comorbidities (Charlson index: 0 [0–1] vs. 1 [0–2]; *p*= 0.04; and age adjusted Charlson: 2 (1–4) vs. 3 (1–4); *p* = 0.009). In terms of urological examination findings, patients in the ABP-CIP group had undergone urological surgical manipulations more frequently in the previous month (2.5% vs. 9.6%; *p* = 0.008) and had a higher prevalence of benign prostatic hyperplasia (26.1% vs. 39.5%; *p* = 0.01). However, both prostatic and urologic tract neoplasia rates were not significantly different between the groups. Clinical characteristics and complementary explorations are summarized in Table 1.

Overall, the rate of *E. coli* resistant to ciprofloxacin in the total cohort (including every ABP-*E. coli*) was 23.1% (n = 96/416), while resistance to cefuroxime was 16.3% (n = 68/416), being consistent with rates reported in Europe. Data on microbiological results is shown in Table 2.

Clinical cure was achieved in 132 episodes (85.7%) in the ABP-CXM group, in front of 158 (96.9%) in the ABP-CIP group (*p* < 0.001). Recurrence was higher in the ABP-CXM group (6.96% (n = 11) versus 0 in the ABP-CIP group (*p* < 0.001)) without differences in reinfection (5 (3.16) vs. 5 (3); *p* = 0.91). No difference was found in prostatic abscess development (1 (0.63) vs. 1 (0.6); *p* = 0.93), nor was mortality (1 (0.63) vs. 1 (0.6); *p* = 16) identified between groups. Data about treatment is summarized in Table 3.

We perform a multivariate analysis of clinical failure to find risk factors, results are in Table 4. Statistical analysis was performed using logistic regression test.

## 3. Discussion

This retrospective observational study suggests that ciprofloxacin may offer superior clinical outcomes compared to high-dose cefuroxime (CXM) in ABP-*E. coli*. Our analysis demonstrates that patients treated with CIP achieved higher clinical cure rates and lower recurrence rates than those receiving CXM. Notably, CIP emerged as a protective factor against clinical failure, highlighting its efficacy as targeted therapy for ABP. These results are particularly compelling given that the CIP group exhibited higher baseline disease severity and more recurrence risk factors, suggesting robust performance in challenging cases. These findings support CIP as the preferred treatment for ABP-*E. coli*, especially in high-risk patients.

Our results are consistent with studies evaluating ABP treatment outcomes. Marquez-Algaba et al. [14] reported relapse rates of 1.8% with CIP versus 3.6% with intravenous beta-lactams (BL), 9.3% with cotrimoxazole (TMP-SMX), and 9.8% with oral BL in their 410-patients ABP retrospective cohort—a pattern that aligns with our observed 7% relapse rate with CXM. In a multivariable analysis, oral BL (OR 5.3, 95% CI 1.2–23.3) and TMP-SMX (OR 4.9, 95% CI 1.1–23.2) were detected as significant relapsed risk factors. Another large retrospective study (n = 33,336 UTI cases) found higher relapse rates with BL (7.3%) and TMP-SMX (5.7%) comparing with FQ (3.9%), with BL emerging as an independent risk factor for relapse (OR 1.81, 95% CI 1.52–2.17) [15]. Madaras-Kelly’s multicenter analysis (n = 73,334) further supported these trends, showing higher clinical failure rates with BL (17.8% in men) versus FQ (12.7%) [16]. Although study designs vary, the collective evidence suggests that oral BL have lower efficacy than fluoroquinolones for male complicated urinary tract infections and acute bacterial prostatitis

Several retrospective studies have assessed oral step-down therapy for Gram-negative bloodstream infections (BSI), though most focus on uncomplicated UTIs or exclude ABP. A 2018 systematic review and meta-analysis of *Enterobacterales* BSI transitioned to oral therapy (n = 2289) found higher recurrence rates with oral BL despite similar mortality [17]. Suboptimal BL dosing was common and may have contributed to this outcome. Although BSI was uncommon in our study, these data support our findings.

Fluoroquinolones are the preferred treatment for ABP in international guidelines, supported by their high intraprostatic penetration (due to their lipophilic nature small molecular size) and concentration-dependent bactericidal activity [7,8]. Although no randomized trials have been conducted, their efficacy is well documented in retrospective studies and clinical experience. However, the IDSA advises against empirical FQ use when *Enterobacterales* resistance exceeds 10%, a threshold commonly surpassed in Europe [18].

Our study reflects real-life clinical practice. Notably, up to two-thirds of patients in the ABP-CXM group had an infection caused by CIP-susceptible strain but were still treated with BL. These patients were likely discharged earlier than those in the ABP-CIP group, as suggested by their shorter length of stay, and microbial susceptibility results may not have been available in time to guide oral therapy decisions. Some authors have proposed systematic urine culture reviews after discharge in patients treated empirically for ABP [3,14]. Given the rising rates of fluoroquinolone resistance, targeted strategies—such as rapid FQ resistance testing or electronic alerts to flag eligible patients—are needed to optimize treatment. In parallel, generating robust evidence through prospective studies to support alternative oral treatment options for ABP remains a key priority.

ABP still lacks standardized diagnostic criteria, which limits diagnostic certainty and comparability across studies. In 2023, Soudais et al. reviewed international guidelines and highlighted inconsistencies in ABP definitions [19]. A recent Delphi consensus proposed diagnostic criteria based on symptoms, systemic signs, pyuria, and microbiological findings [20]. Still, a unified international definition is needed. Further complicating this, clinical presentation varies with age and care setting. A 2019 study by Etienne et al. (n = 586) found that older patients (median age 84) had fewer typical symptoms (63% fever, 50% urinary symptoms), and treatment approaches differed across departments despite similar outcomes [5]. In our cohort, 96% of patients had fever at presentation, supporting appropriate case selection.

Serum prostate-specific antigen (sPSA) is a potentially useful marker for ABP diagnosis but remains underexplored. Prospective studies have shown that 81–83% of men with febrile UTI present with elevated sPSA levels that normalize after treatment [21,22]. In a recent study by our group, 89% of episodes in men with febrile UTI had sPSA > 5 ng/mL (median 16.7 ng/mL) [23]. These findings suggest that sPSA may reflect prostate involvement in male UTIs and could help guide treatment decisions.

Several risk factors for poor ABP outcomes have been described. Two large retrospective studies (n = 130 and n = 614) reported higher rates of abscesses and recurrence in patients with prior urological manipulation [24,25]. Another study (n = 437) found progression to chronic prostatitis or pelvic pain syndrome in 11.8% of cases, associated with alcohol use, diabetes, voiding symptoms, prior manipulations, prostate enlargement, catheterization, and shorter antibiotic courses, regardless of antibiotic type [26]. In our cohort, prior manipulation was not associated with higher rates of clinical failure or abscesses. However, these patients were overrepresented in the ABP-CIP group, suggesting that ciprofloxacin may have reduced the risk of complications. In multivariate analysis, ciprofloxacin use was a protective factor against recurrence, while the history of UTI in the previous year was associated with clinical failure. This may reflect underlying chronic prostatitis, which often presents as recurrent lower UTI and may require prolonged treatment [2,11].

Our study has several limitations. First, it was conducted at a single center which may limit the generalizability of the findings. Second, the long study period could have introduced variability in clinical practice. Third, its retrospective design inherently limits data consistency and completeness. Finally, most patients were diagnosed in the emergency department, where variability in physician experience and decision-making may have influenced diagnostic and treatment approaches.

Nonetheless, the study also has notable strengths. It includes a well-characterized cohort with comprehensive clinical data. Patients with catheter-related UTIs were excluded, reducing the likelihood of misclassification. Moreover, by focusing solely on patients treated with either cefuroxime or ciprofloxacin, the analysis allows comparison between these two therapeutic options.

## 4. Material and Methods

We conducted a retrospective, observational study at a 400-bed teaching hospital in Barcelona, Spain (Hospital Universitari Mutua Terrassa), including all adult patients (>18 years) diagnosed with ABP-*E. coli* between January 2010 to October 2023. Eligible patients were treated with high-dose cefuroxime (500 mg every 8 h) or standard-dose ciprofloxacin (500 mg every 12 h) and were diagnosed at the Emergency Department or during hospitalization. Patients were followed up to 90 days after the initial episode.

We reviewed all ABP cases in the hospital database and included those presenting with symptoms consistent with ABP and a positive urine culture showing significant growth of *E. coli* (>100,000 CFU). Data were collected from medical and nursing records, including clinical presentation, physical examinations findings, laboratory and microbiological results, acquisition site, imaging studies, treatment regimens, adverse effects, and clinical outcomes, as well as age and comorbidities present in the year prior to infection.

Informed consent was waived due to the retrospective nature of the study. According to institutional and national regulations, formal ethics committee approval was not required.

Exclusion criteria, variables definitions, and statistical methods are detailed in Appendix A.

## 5. Conclusions

In conclusion, our findings support ciprofloxacin as the preferred treatment for *E. coli*-related ABP, particularly in patients at higher recurrence risk. The history of previous UTIs emerged as a significant predictor of clinical failure, underscoring the need for closer follow-up and potentially alternative therapeutic strategies. Notably, many patients infected with ciprofloxacin-susceptible strains were treated with beta-lactams. This highlights the challenge of managing community-acquired infections requiring prolonged therapy, where empirical decisions are often made without susceptibility data. A more personalized approach—considering individual risk factor and rapid diagnostics—is needed. In the context of increasing resistance to quinolones, it is essential to produce prospective studies and explore effective oral alternatives such as fosfomycin.

## Figures and Tables

**Figure 1 antibiotics-14-00681-f001:**
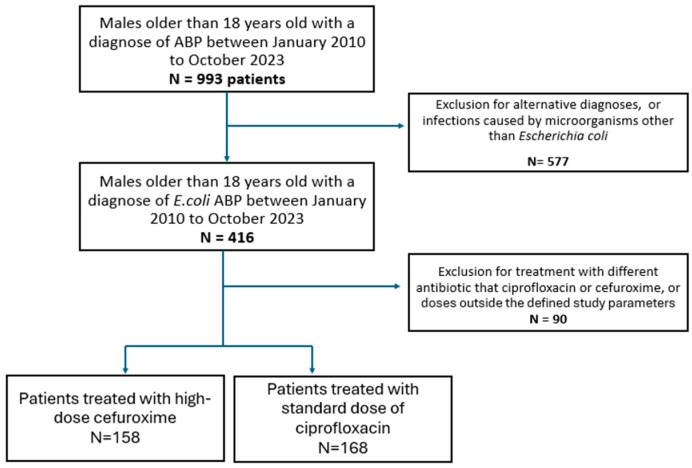
Patient selection flow.

**Table 1 antibiotics-14-00681-t001:** Clinical characteristics and complementary examinations.

	ABP-CXM (n = 158)	ABP-CIP (n = 168)	*p*
Clinical presentation *			
Time until consultation (days)	1 (1–3)	1 (1–3)	0.72
Fever at consultation	155 (98.1)	158 (94.1)	0.06
Days of fever	1 (1–2)	1 (1–2)	0.71
Dysuria	134 (84.8)	145 (86.8)	0.6
Urinary frequency	108 (68.8)	116 (69.4)	1
Urinary urgency	24 (15.3)	38 (23)	0.21
Bladder urgency	69 (43.7)	58 (34.7)	0.08
Suprapubic pain	34 (21.5)	33 (20)	0.35
Perineal pain	17 (10.8)	13 (7.8)	0.26
Inguinal pain	6 (3.8)	8 (4.7)	0.35
Acute urinary retention	10 (6.37)	16 (9.6)	0.28
Hematuria	14 (8.9)	8 (4.76)	0.14
Shock	3 (1.9)	8 (4.76)	0.15
Diagnostic test			
Digital rectal examination performed	33 (20.9)	46 (27.4)	0.2
Painful examination	26 (16.7)	31 (18.5)	0.48
Echography performed	25 (15.8)	54 (32.1)	0.001
Anormal echography	4 (16)	8 (14.8)	0.9
CT perform	19 (12)	34 (20.2)	0.05
CT abnormalities	4 (21.05)	13 (38.2)	0.2
PSA performed	34 (21.5)	39 (23.2)	0.71
PSA total (ng/mL)	16.7 (9.2–28)	14.2 (7.47–24.5)	0.77
CRP performed	155 (98.1)	168 (100)	0.06
CRP value (mg/L)	113 (55–184)	112.5 (42.2–175.7)	0.54
Type of infection			
Community-acquired	146 (92.4)	142 (84.5)	0.07
Healthcare related	9 (5.7)	22 (13.1)	
Hospital acquired	3 (1.9)	4 (2.4)	

* Values are expressed as n (%) or median (interquartile range), as appropriate. ABP-CXM: *E. coli* acute bacterial prostatitis treated with high-dose cefuroxime, ABP-CIP: *E. coli* acute bacterial prostatitis treated with ciprofloxacin.

**Table 2 antibiotics-14-00681-t002:** Microbiological findings.

	ABP-CXM (n= 158)	ABP-CIP (n = 168)	*p*
Microbiology N (%)			
Bacteremia	14 (10.1)	52 (33.3)	<0.001
Antimicrobial resistance			
EBSL *	1 (1.36)	3 (1.78)	0.6
Resistant CIP	36 (22.8)	1 (0.6)	<0.01
Resistant A/C	42 (26.6)	44 (26.2)	0.93
Resistant CXM	5 (3.16)	8 (4.76)	0.58
Resistant TMP-SMX	27 (17.1)	26 (15.5)	0.74
MultiR #	20 (12.7)	12 (7.14)	0.09

* EBSL: Extended-Spectrum Beta-Lactamases, # Magiorakos’s definition of multirresistance. ABP-CXM: ABP-*E. coli*, treated with high-dose cefuroxime, ABP-CIP: ABP-*E. coli*, treated with ciprofloxacin.

**Table 3 antibiotics-14-00681-t003:** Treatment.

	ABP-CXM (n = 158)	ABP-CIP (n = 168)	*p*
Treatment *			
Hospitalization	76 (48.4)	148 (88.6)	<0.001
Admission days	2 (1–4.75)	3 (2–5)	0.002
Empirical treatment with in vitro activity	152 (96.2)	163 (97)	0.68
Empirical treatment			
- 3rd generation cephalosporins	146 (92.4)	124 (73.8)	<0.001
- Monotherapy #	153 (96.8)	153 (91.1)	0.03
Treatment duration. Days	21 (21–22)	21 (19–21)	<0.001
Oral treatment duration. Days	20 (17–21)	17 (14–19)	<0.001

* Values are expressed as n (%) or median (interquartile range), as appropriate. # Any single antibiotic was started as empirical treatment. ABP-CXM: ABP-*E. coli*, treated with high-dose cefuroxime, ABP-CIP: ABP-*E. coli*, treated with ciprofloxacin.

**Table 4 antibiotics-14-00681-t004:** Multivariable analysis.

	OR	*p* Value	95% IC
Use of ciprofloxacin as sequential oral treatment	0.17	0.001	0.06–0.47
UTI * the previous year	2.52	0.03	1.12–5.74
Days of fever until consultation	1.28	0.10	0.95–1.72

* Urinary tract infection

## Data Availability

The original contributions presented in this study are included in the article. Further inquiries can be directed at the corresponding author.

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
