# Peer review of "Clinical Outcomes of Escherichia coli Acute Bacterial Prostatitis: A Comparative Study of Oral Sequential Therapy with β-Lactam Versus Quinolone Antibiotics"

_antibiotics, 2025, doi:10.3390/antibiotics14070681_

Round 1

Reviewer 1 Report

Comments and Suggestions for Authors

Study outlines the Ciprofloxacin is preferred over cefuroxime for treating ABP-E.coli patients with a higher risk of recurrence, particularly when prior UTIs  are present, as it reduces recurrences and mortality. This study is very interesting and provides highly useful information for treatment strategies related to ABP-E. coli. I support the manuscript for publication and have added some comments to improve it. 

Comments: 

  1. The discussion section is too long; the authors could condense it for clarity.  
  2. To enhance broad audience understanding, the authors could add "antibiotics" to the end of the title “Clinical Outcomes of Escherichia coli Acute Bacterial Prostatitis: A Comparative Study of Oral Sequential Therapy with β-Lactam versus Quinolone antibiotics”
  3. Authors could mention what demographic information was collected about the patients in the methods section. 
  4. How were baseline E. coli levels determined in the study?
  5. Abstract section need to be corrected for clarity
  6. What is the treatment strategy used for patients with ABEC who have higher risk of recurrence when comparing high-dose cefuroxime (500 mg every 8 hours) vs standard-dose ciprofloxacin (500 mg every 12 hours), why is there difference high dose vs standard dose
  7. Line 106,  "In terms of urological conditions..." can be  "In terms of urological examination findings..." for clarity. 
  8. Line 79, “we included 326 episodes” can be clearly mention of what episodes
  9. Figure 1 legend needs improvement
  10. The study found ciprofloxacin to be a protective factor against clinical failure. What mechanisms  might explain its superior efficacy compared to cefuroxime could discuss in discussion section 

Author Response

Please see the attachament. 

The English of the entire article has been improved

Reviewer 2 Report

Comments and Suggestions for Authors

The study addresses a highly relevant clinical question—the optimal antibiotic choice for E. coli-associated acute bacterial prostatitis (ABP)—with practical implications for antimicrobial stewardship.

Suggestion for Improvement;

Include antibiotic susceptibility data for E. coli isolates to contextualize the results.

Report duration of therapy and whether patients were treated as inpatients or outpatients.

Add a Kaplan-Meier curve for time to recurrence or failure to enhance visual interpretation.

Consider stratifying results based on complicated vs. uncomplicated ABP cases.

Introduction;

The Introduction provides a solid foundation for the study and aligns with research reporting norms. With a few grammar fixes and slight content refinement (especially citation consistency and data context), it can be improved for clarity and professional polish.

Methods;

Add ethics committee approval reference number if required by your target journal.

Mention software used for statistical analysis (e.g., SPSS, R) in the appendix or methods.

Include exclusion criteria, if any (e.g., patients with polymicrobial cultures or incomplete records).

If you did sample size calculation or power analysis, consider briefly noting it here or in Appendix A.

Result;

Double-check figure/table placements in the final layout.

Consider plotting key outcome differences (e.g., cure rate, recurrence) using bar graphs or forest plots.

Clearly state missing data handling (e.g., number of patients with imaging/PSA not done).

Include a summary sentence after each table/figure for emphasis.

Discussion:

  • The discussion is too long (over 130 lines), and several concepts are repeated (e.g., fluoroquinolone resistance, BSI recurrence, risk factors).
  • Tighten the prose. Condense overlapping studies and remove tangents (e.g., detailed sPSA discussions could be summarized or moved to Results or Appendix).
  • The narrative occasionally shifts abruptly between topics (e.g., from ABP treatment to bloodstream infections, then back to prostate-specific antigen).
    • Clinical Efficacy of CIP vs. CFX
    • Comparison with Prior Studies
    • Resistance and Empirical Treatment Challenges
    • Diagnostic Limitations
    • Study Strengths and Limitations
    • Implications for Future Practice
  • The discussion frequently references retrospective studies, which may weaken the perceived strength of conclusions.
  • Acknowledge this limitation explicitly and emphasize the need for prospective trials.
  • Standardize all abbreviations (define once early in Methods or Discussion and use consistently).
  • Awkward constructions (e.g., “being those differences higher in the men subgroup” → “these differences were more pronounced in men”).
  • Unnecessary passive voice (e.g., “was found” → “we found”).
  • Missing articles (e.g., “treatment with CIP was protective” → “a protective treatment option”).
  • The sPSA portion is lengthy and detracts from the main narrative.
  • Summarize in 3–4 lines and cite studies without excessive detail.

Conclusion;

  • From main findings → implications → future directions.
  • Refined language for clarity and scientific professionalism.

Comments on the Quality of English Language

Need to Improved

Author Response

The English of the entire article has been improved

Reviewer 3 Report

Comments and Suggestions for Authors

Dear authors,

I enjoyed reading your paper but I have few issues that I would like you to consider: 

1. First, I have an issue with the statement that European Centre for Disease Prevention and Control reported Escherichia coli fluroquinolones resistance at 23% across Europe in 2024 and the study was focused on analysing data from 2010-2023, in terms that the resistance level was not that high in that period. Maybe that was one of the factors responsible for better outcomes, and maybe not. Authors might provide E.coli resistance level for the study period, and not only to fluroquinolones but for cefuroxime as well. 

2. Also, cefuroxime does not have labeled indication for prostatitis, I am not sure it is scientifically supported to treat prostatitis using this antibiotic. Anyhow it is an off-label use.

3. Why did you define dose of cefuroxime 500 mg p.o. every 8 hours as high dose for hospital patients since cefuroxime can be dosed 1500 mg i.v. every 8 hours?

4. What were other factors included in multivariate analysis? You have not explained it clearly in methodology or results sections.

5. It is written that bacteraemia was detected in 66 patients, what is the source of other isolated E.coli? 

6. You refer to usage of beta lactams if all of them have the same effect, third generation of cephalosporins is more suitable for prostatitis then second, not to mention carbapenems etc.

Reviewer 4 Report

Comments and Suggestions for Authors

Dear Authors,

I have reviewed your article titled “Clinical Outcomes of Escherichia coli Acute Bacterial Prostatitis: A Comparative Study of Oral Sequential Therapy with β-Lactam versus Quinolone.” First of all, thank you for your efforts. It is generally well designed and written. Some revisions need to be made before publication:

- There are places where the bacterial names are not written in italics. ‘Enterobacterales’ is written as ‘Enterobacteriaceae’ in some places with the old nomenclature. This should be corrected.

- CXM is written in some places, CFX in some places. A single abbreviation should be used.

- In Figure 1, the CIP group is stated as 158 people, but it should be 168.

- I think there is a mistake in Table 3. Why was there a significant difference (p<0.001) despite the equal treatment duration (21 days)? In addition, what is meant by monotherapy should be stated. Again, the explanation of the ‘*’ symbol should be given below the table.

- Line 135-138, “We perform a multivariate analysis of clinical failure to found risk factors. We found 135 that treatment with ciprofloxacin (OR: 0.16, CI 0.06-0.47, p <0.001) was a protective factor, whereas a UTI in the previous year (OR: 2.56, CI 1.12-5.75; p=0.025) was a risk factor for clinical failure. Statistical analysis was performed using logistic regression test.” It is recommended that this information be given in the table as well.

- Line 256, “In a recent study by our group, we evaluated recurrence rates based on different antibiotic durations in men with febrile UTI (31).” I think the 31st reference here will be 29.

Best regards,

Round 2

Reviewer 3 Report

Comments and Suggestions for Authors

Dear authors,

Thank you for your responses. I am satisfied with your explanations.

One remaining question I have is why you initially considered using cefuroxime as part of empirical treatment. The result of your study is quite expected, since cefuroxime has much lower potential to treat bacterial prostatitis due to its pharmacological and pharmacokinetic properties. Additionally, it is not listed as a therapeutic option in the guidelines you referenced. I understand that it is recommended by your local guidelines, however it does not add anything new to our previous knowledge of bacterial prostatitis treatment.